# Improved genome recovery and integrated cell-size analyses of individual uncultured microbial cells and viral particles

Ramunas Stepanauskas [1], Elizabeth A. Fergusson[1], Joseph Brown[1], Nicole J. Poulton[1], Ben Tupper[1], Jessica M. Labonté[1,3], Eric D. Becraft[1], Julia M. Brown[1], Maria G. Pachiadaki[1], Tadas Povilaitis[2], Brian P. Thompson[1], Corianna J. Mascena[1], Wendy K. Bellows[1] & Arvydas Lubys[2]

Microbial single-cell genomics can be used to provide insights into the metabolic potential, interactions, and evolution of uncultured microorganisms. Here we present WGA-X, a method based on multiple displacement amplification of DNA that utilizes a thermostable mutant of the phi29 polymerase. WGA-X enhances genome recovery from individual microbial cells and viral particles while maintaining ease of use and scalability. The greatest improvements are observed when amplifying high G+C content templates, such as those belonging to the predominant bacteria in agricultural soils. By integrating WGA-X with calibrated index-cell sorting and high-throughput genomic sequencing, we are able to analyze genomic sequences and cell sizes of hundreds of individual, uncultured bacteria, archaea, protists, and viral particles, obtained directly from marine and soil samples, in a single experiment. This approach may find diverse applications in microbiology and in biomedical and forensic studies of humans and other multicellular organisms.

[1] Bigelow Laboratory for Ocean Sciences, 60 Bigelow Drive, East Boothbay, Maine 04544, USA. [2] Thermo Fisher Scientific Baltics, Graiciuno 8, LT-02241 Vilnius, Lithuania. [3] Present address: Department of Marine Biology, Texas A&M University at Galveston, Galveston, Texas 77553, USA. Correspondence and requests for materials should be addressed to R.S. (email: rstepanauskas@bigelow.org)

Single-cell genomics (SCG) retrieves information-rich genomic blueprints from the most fundamental units of life[1–6]. This is particularly significant in the case of bacteria, archaea, and protists, where individual cells constitute complete organisms. Such unicellular individuals comprise the vast majority of biological diversity on our planet, with recent estimates indicating over a trillion species[7]. Only a small fraction of them are amenable to the cultivation-based, classical microbiology studies[8]. SCG[9–12] as well as the assembly and binning of metagenomic sequences[13–16] are instrumental in the deciphering of the biological features of many deep branches of the tree of life that constitute a significant fraction of our planets biota yet remained unknown to science until recently. In addition, due to its ability to retrieve genetic information from all DNA molecules in a cell, SCG opens a window of opportunity to study microbial physical interactions, such as infections, symbioses, and predation, directly in their natural environment[12, 17–19]. Finally, by circumventing the need for arbitrary taxonomic binning, as in the case of metagenomic assemblies, SCG improves our understanding of microbial microevolutionary processes[20, 21] and helps calibrate the performance and interpretation of community omics tools[17, 22]. A major, still underutilized opportunity lies in the integration of SCG with single-cell phenotype analyses, which can provide deeper insights into the roles of uncultured microbial groups in nature and inform their practical utilization in biotechnology[23].

The SCG workflow generally involves individual cell isolation and lysis, genomic DNA (gDNA) amplification and sequencing, and computational sequence analyses[1–6]. Most cells contain only one or a few copies of their genome, constituting femtograms to picograms of DNA, which is not sufficient for direct analysis with current sequencing technologies[24]. Therefore, gDNA amplification is essential in the SCG workflow. Since its invention in 2002[25], multiple displacement amplification (MDA) has been the most widely used gDNA amplification method in SCG due to its multiple advantages: (a) long, overlapping amplicons that are well suited for genomic sequencing and subsequent de novo assembly; (b) high fidelity of the phi29 polymerase; and (c) simple reaction setup that reduces the risk of handling errors and contamination and facilitates automation. However, single-cell MDA exhibits significant limitations, such as incomplete and uneven genome amplification, potential biases against high G+C templates, and chimera formation[26–28]. For example, even a 1000× sequencing depth typically recovers only an average of <50% of the genome from individual microbial cells[10, 29]. Several studies report reduced amplification biases through modified methods such as performing MDA in nano-liter-scale and pico-liter-scale liquid volumes[30–32] or in agarose gels[33], or employing protein priming[34]. However, these approaches do not address the systemic MDA bias against high %GC templates and remain difficult to integrate into high-throughput workflows that involve sorting of specific cell types or single-cell phenotype analyses. The

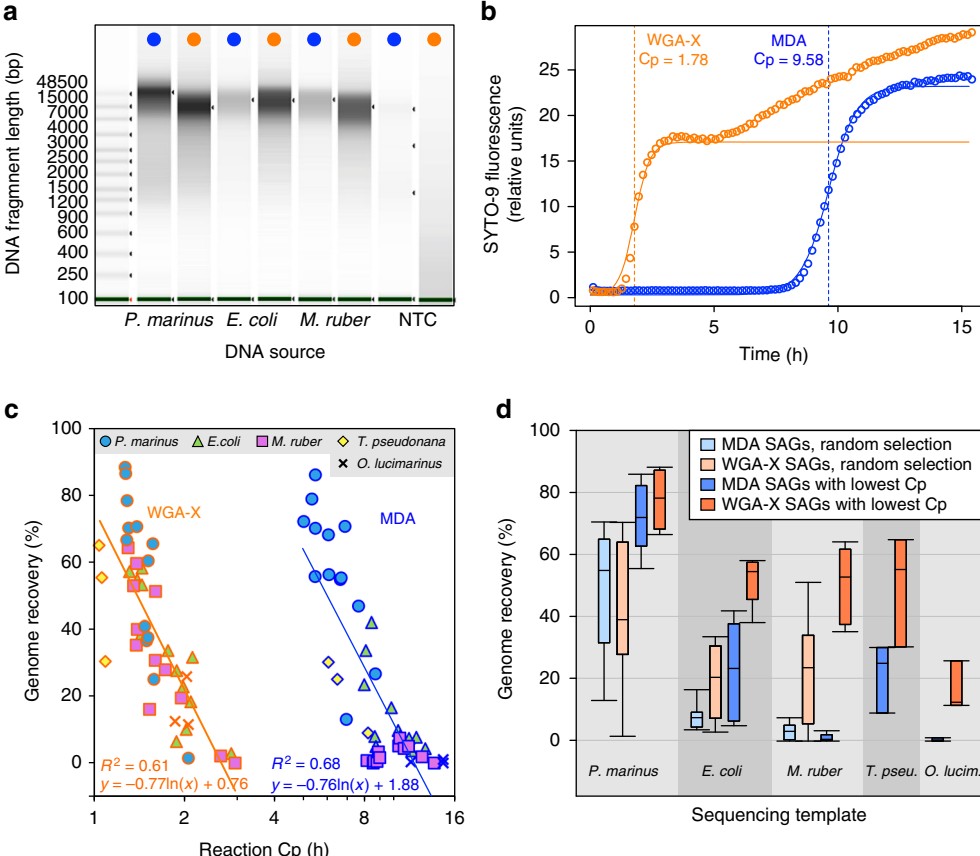

**Fig. 1** Comparison of WGA-X and MDA performance with microbial benchmark strains. **a** Electrophoresis gel images of WGA-X (*orange dots*) and MDA (*blue dots*) products obtained from three bacterial benchmark strains and from no-template negative controls (*NTC*). **b** Examples of WGA-X and MDA reaction kinetics, where reaction critical point (*Cp*) is estimated as the time required to reach the inflection point of the reaction's exponential phase. **c** Correlation between reaction Cp and genome recovery from SAGs. **d** Average, standard deviation, and range of genome recovery from SAGs of the five benchmark strains, where SAGs were selected either at random or based on their lowest Cp values. In **c** and **d**, each bacterial strain data set derives from eight randomly selected SAGs and five SAGs with the lowest Cp; each eukaryote data set derives from three SAGs with the lowest Cp. Assemblies of bacterial and eukaryote SAGs were produced from five million and twenty million of 2 × 150 bp reads, respectively

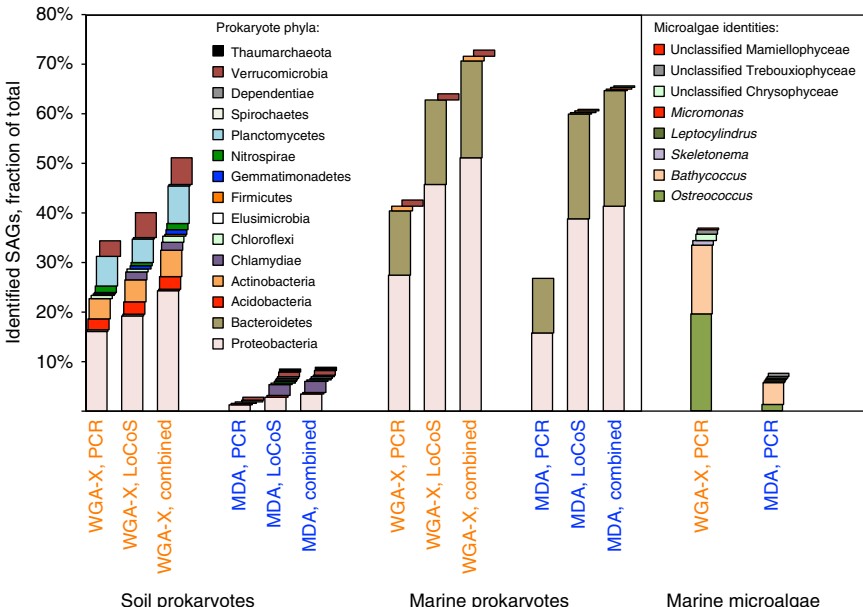

**Fig. 2** Taxonomic assignments of environmental microbial SAGs. The following approaches were used: PCR-based sequencing of SSU rRNA genes followed by classification with CREST (prokaryotes) or Silva Incremental Aligner (microalgae); LoCoS followed by CheckM, Metaxa, and CREST; and **a** combination of the two approaches. 317 SAGs from each environment, cell type, and gDNA amplification method were analyzed

alternative methods PicoPLEX and MALBAC, which combine isothermal and polymerase chain reaction (PCR) steps, were shown to increase the evenness of single-cell gDNA amplification as compared with the MDA in studies of human single cells[35, 36]. Unfortunately, both PicoPLEX and MALBAC are susceptible to contamination with microbial DNA and high error rates[4, 37], whereas their multi-step setup and thermal cycling requirements hamper scalability and automation. Thus, limitations of current gDNA amplification methods remain among the key challenges for SCG.

Here we present WGA-X, an MDA-like method that utilizes a thermostable mutant of the phi29 polymerase[38]. Using benchmark strains and environmental samples, we demonstrate that WGA-X enhances genome recovery from individual microbial cells and viral particles while retaining MDA's ease of use and scalability. The greatest improvements were observed when amplifying high G+C content templates, such as the predominant bacteria of an agricultural soil sample. By integrating WGA-X with high-throughput genomic sequencing and index fluorescence-activated cell sorting (FACS), we were able to analyze genomic sequences and cell sizes of hundreds of individual, uncultured bacteria, archaea, protists, and viruses that comprise complex marine and soil microbiomes.

## Results

**WGA-X evaluation with benchmark microbial cultures**. Optimized WGA-X and standard MDA were used to generate single amplified genomes (SAGs) of three previously sequenced strains of bacteria and two strains of eukaryotic algae spanning diverse G+C content (36–63%), genome complexity (2–32 Mbp in 1–32 chromosomes), and phylogenetic affiliations (Cyanobacteria, Proteobacteria, Deinococcus-Thermus, Heterokontophyta, and Chlorophyta; Supplementary Table 1). The majority of amplicons of both MDA and WGA-X were >7 kbp in length (Fig. 1a). Over 99% of the no-template negative control MDA products contained no detectable DNA. Over 50% of no-template WGA-X products contained low levels of low molecular weight DNA that did not form electrophoresis bands and did not produce >2 kbp

contigs in de novo assemblies from their shotgun reads. After completion of single-cell gDNA amplification, the final DNA amount in 10 µl WGA-X and MDA reactions was $1087 \pm 475$ and $309 \pm 296$ ng, respectively. Single-cell WGA-X reactions were significantly faster than MDA reactions containing the same templates, with an average critical point (Cp; the time required for reaching the inflection point of reaction's exponential phase) of 1.9 vs. 9.6 h (Fig. 1b, c; $p < 0.001$, Student $t$-test).

We generated 317 SAGs for each benchmark bacterial strain (Supplementary Table 1) and gDNA amplification method, and randomly selected 8 SAGs of each treatment for genomic sequencing. On the basis of prior[17] and current (Fig. 1c) observations of SAG genome recovery correlating negatively with high MDA Cp values, we also sequenced five bacterial and three eukaryote SAGs of each strain and gDNA amplification method, selected based on lowest Cp values. We found that WGA-X SAGs had better genome coverage by raw reads (Supplementary Table 2) and larger de novo assemblies compared to MDA SAGs ($p < 0.05$; Student $t$-test; Fig. 1d), with the exception of *Prochlorococcus marinus* (benchmark culture with the lowest G+C content). The use of WGA-X Cp as a selection criterion further increased the assembly size, compared to a random SAG selection ($p < 0.05$). When comparing SAGs with the lowest Cp values, the difference in de novo genome assembly size in WGA-X vs. MDA SAGs was 2.3× for *E. coli*, 55× for *M. ruber*, 1.4× for *T. pseudonana*, and 35× for *O. lucimarinus* (Fig. 1d).

We found no differences between WGA-X and MDA in the fraction of reads mapping to the reference genomes and the frequency of read-level chimeras, single nucleotide polymorphisms (SNPs) or insertions, whereas the frequency of deletions was slightly higher in WGA-X reads (Supplementary Table 2). Likewise, there were no consistent differences between the two gDNA amplification methods in the frequency of misassemblies, mismatches, and indels in de novo assemblies (Supplementary Fig. 1). SAGs of *P. marinus* tended to have the most contiguous assemblies with fewest misassemblies, mismatches, and indels, likely owing to the small genome size and the low number of repeat regions. Notably, the estimated frequency of mismatches and indels in assemblies of *T. pseudonana* SAGs was an order of

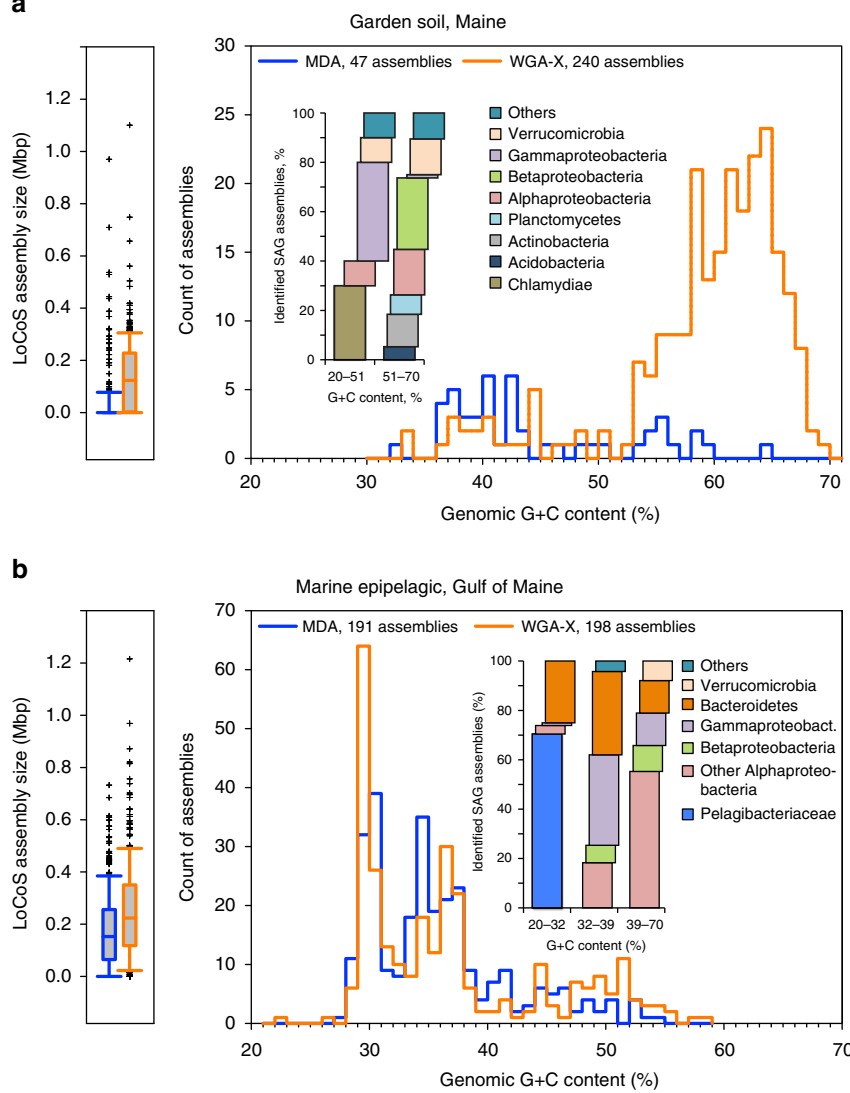

**Fig. 3** Results of low-coverage sequencing (*LoCoS*) of WGA-X and MDA SAGs of prokaryotes from **a** garden soil and **b** coastal ocean. Presented are de novo assembly sizes, G+C content, and phylogenomic assignments. A total of 317 SAGs were generated from each environment using each gDNA amplification method. The count of successful SAG assemblies is provided next to the gDNA amplification method. *Insets* indicate the phylogenomic assignments of SAGs within discernable G+C intervals

magnitude higher than in SAGs of other benchmark strains, likely because only half of the chromosomes of this diploid and highly polymorphic organism are present in the reference assembly[39]. Only assemblies of *T. pseudonana* and *O. lucimarinus* SAGs contained regions that did not align to published reference genomes. These regions were dominated by plastid, mitochondrial, and chromosomal genes with best BLAST hits to closely related organisms, indicating that these regions were probably indigenous to the analyzed strains but were missed in reference assemblies obtained using conventional techniques (Supplementary Fig. 2).

**SCG of soil and marine microbiomes**. To compare the performance of the two gDNA amplification methods in environmental samples, we generated 317 WGA-X SAGs and 317 MDA SAGs of each of the following: garden soil prokaryotes, surface ocean prokaryotes, and surface ocean microalgae. The PCR and subsequent sequencing of the small subunit (SSU) rRNA gene identified a larger fraction of WGA-X SAGs than MDA SAGs: 34

vs. 3% soil prokaryotes, 43 vs. 27% marine prokaryotes, and 37 vs. 8% marine microalgae (Fig. 2). This was accompanied by pronounced compositional differences: compared to the WGA-X SAG libraries of prokaryotes, MDA SAGs were underrepresented in Acidobacteria, Actinobacteria, Chloroflexi, Gemmatimonadetes, Planctomycetes, and Verrucomicrobia.

To perform PCR-independent SAG identification and to obtain genomic information from each SAG at a minimal cost, we performed low-coverage genomic sequencing (LoCoS) and subsequent phylogenomic analyses on all prokaryote SAGs (see Methods). LoCoS assemblies consisting of at least one >2 kb contig were obtained from 240 (76%) soil WGA-X SAGs and 47 (15%) soil MDA SAGs, indicating that WGA-X introduced a 5× improvement. For the marine sample, 198 (62%) WGA-X SAG and 191 (60%) MDA SAG assemblies consisting of at least one >2 kb contig were obtained. LoCoS assemblies of WGA-X SAGs were significantly larger than MDA SAG assemblies (*p* < 0.001; Student *t*-test) in both soil (144 vs. 30 kbp average size) and marine (244 vs. 179 kbp) samples (Fig. 3). LoCoS improved SAG identification and, when combined with the PCR-based approach,

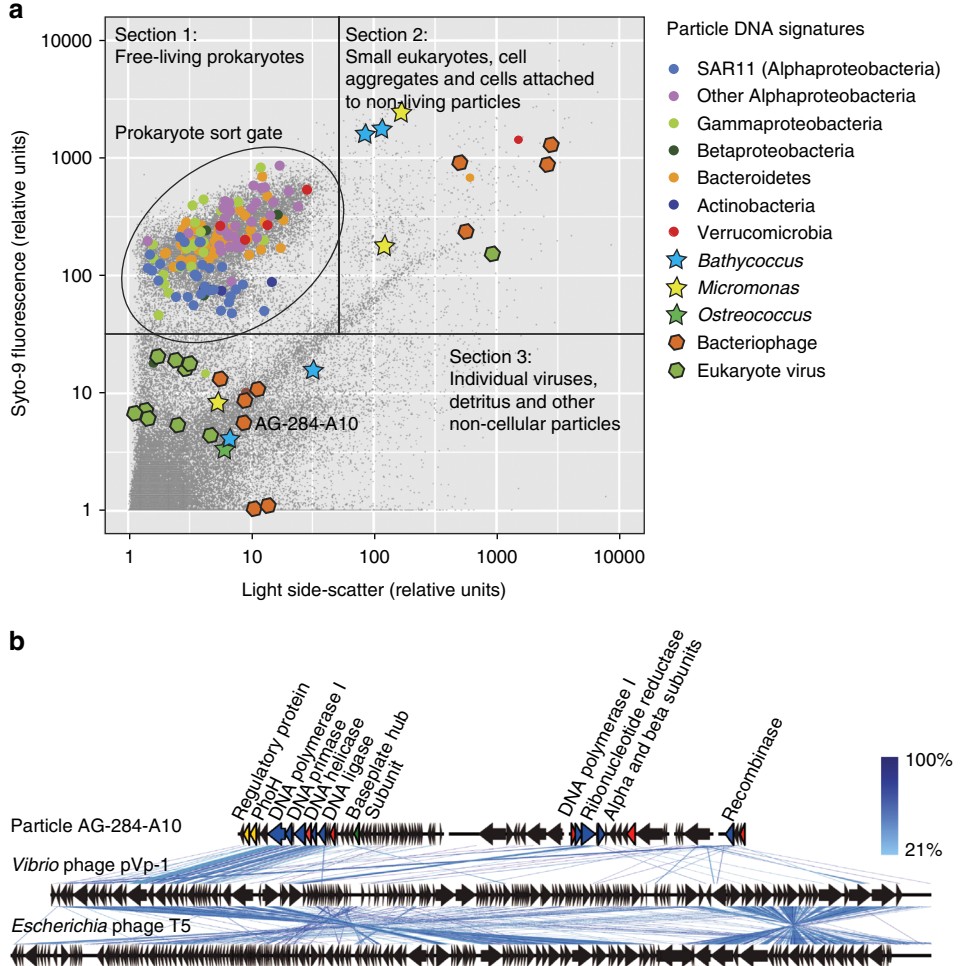

**Fig. 4** Integration of index FACS and single-cell genomics. **a** Flow cytometry optical properties of individual marine prokaryotes and non-cellular particles from which SAGs were generated. Section boundaries indicate tentative separation of particle types. **b** Genome alignments of the individual viral particle AG-284-A10 and its closest sequenced relatives *Siphoviridae Vibrio* phage pVp-1 (Genbank #JQ340389) and *Escherichia* phage T5 (Genbank AY543070). Each *arrow* represents a gene in the direction of transcription. *Arrows* represent genes related to regulation (*yellow*), DNA replication (*blue*), structure (*green*), unknown phage proteins (*red*), and hypothetical phage proteins (*gray*). The *scale bar* in **b** indicates peptide-sequence identity

increased the fraction of identified WGA-X SAGs of soil and marine prokaryotes to 51 and 73%, respectively (Fig. 2). Notably, LoCoS but not PCR detected Chlamydiae SAGs (Fig. 2; Supplementary Data 1). Furthermore, LoCoS revealed infections by novel phages in 22 WGA-X SAGs and 9 MDA SAGs (Supplementary Data 2).

The two gDNA amplification methods showed major differences in the G+C content of soil LoCoS assemblies (Fig. 3a). Although the majority of WGA-X SAGs had G+C >50%, only a few MDA SAG assemblies were in this G+C range, resulting in divergent average G+C content of WGA-X SAGs (59%) vs. MDA SAGs (45%). The G+C content of marine prokaryote SAGs averaged 36% for both WGA-X and MDA libraries (Fig. 3b). The G+C distribution was bi-modal in soil SAG libraries and tri-modal in marine SAG libraries. In soil, SAGs with G+C <50% were dominated by Gammaproteobacteria and Chlamydiae, whereas SAGs with G+C >50% were enriched in Betaproteobacteria, Alphaproteobacteria, Acidobacteria, Actinobacteria, Chloroflexi, Gemmatimonadetes, Planctomycetes, and Verrucomicrobia. In the marine sample, SAGs with G+C <32% were dominated by the SAR11 clade (Alphaproteobacteria) and Bacteroidetes; SAGs with G+C from 32 to 39% were dominated by Gammaproteobacteria, Bacteroidetes, and non-SAR11 Alphaproteobacteria; whereas SAGs with G+C >39% were enriched in

non-SAR11 Alphaproteobacteria, predominantly from the families Rhodobacteriaceae and Rhodospirillaceae.

**Integration of genomic and cell-size properties of individual cells.** Taking advantage of the index sort capability of the BD InFlux Mariner cell sorter, we recorded light scatter and fluorescence intensities of each individual particle that was deposited into a specific microplate well. As only one particle was deposited per well, we could directly link these FACS data and genomic sequences of the same particle. The application of this approach demonstrated phenotypic differences among certain phylogenetic groups of marine bacteria (Fig. 4a) and microalgae (Supplementary Fig. 3). For example, cells of the abundant Alphaproteobacteria group SAR11 had low SYTO-9 fluorescence, indicative of their low nucleic acid content, as compared to most other bacterial groups (Fig. 4a).

To transform the relative units of light forward scatter to the absolute units of cell diameter equivalents, we calibrated the cell sorter with microbial strains of known cell size (Fig. 5a). The application of this technique on soil prokaryotes enabled us to determine sizes of the same individual, uncultured cells that were also analyzed genomically, providing an average diameter equivalent of 0.65 µm (Supplementary Data 1). There was a

general consistency in sizes among closely related cells (e.g., within the AKIW543 cluster of Actinobacteria), whereas some of the higher-level phylogenetic groups contained a wide range of estimated cell sizes (Fig. 5b; Supplementary Data 1). Among phyla with multiple representatives, Thaumarchaea and Nitrospirae cells were the smallest, with the average cell diameter

equivalents of ~0.2 µm. In contrast, average diameter equivalents of Acidobacteria, Chloroflexi, and Planctomycetes cells were largest and exceeded 1 µm.

**Individual viral particles and other extracellular genetic elements.** To explore the composition of marine non-cellular particles, we generated 317 WGA-X SAGs and 317 MDA SAGs from particles that fell outside of the typical FACS gate for free-living prokaryotes (Fig. 4a). All wells were sequenced using the LoCoS approach and eleven WGA-X SAGs with the lowest Cp values were sequenced deeper. This resulted in 36 assemblies of WGA-X SAGs and 4 assemblies of MDA SAGs that were >5 kbp. The combination of genome content and optical properties of these 40 particles, enabled by index FACS, suggested the following identities: 9 eukaryotic viruses, 7 bacteriophages, 4 unicellular algae, 2 particle-attached bacteria or bacterial colonies, 4 infected cells, 5 particles of cellular debris, and 7 particles that could not be identified due to their limited homology to DNA sequences in public databases (Fig. 4a; Supplementary Data 1). Several particles containing bacterial and viral DNA had higher light scatter than microalgae, suggesting that these were aggregates of infected cells or viruses attached to non-living material. A comparative genome analysis of the phage-like particle AG-284-A10 showed similarity to the *Siphoviridae vibrio* phage pVp-1 and *Escherichia* phage T5 (Fig. 4b). Two other virus-like SAGs, AG-284-B08 and AG-284-K02 were most closely related to BpV1 and BpV2 viruses of the unicellular green alga *Bathycoccus* (Supplementary Fig. 4A). SAGs AG-284-A05 and AG-284-J02 were distantly related to the large phycodnaviruses "1" and "2" (Supplementary Fig. 4B).

## Discussion

We have shown here that, in comparison with conventional MDA, the use of WGA-X provides better genome recovery while maintaining similar fidelity for bacterial benchmark strains (Fig. 1c, d; Supplementary Fig. 1; Supplementary Tables 1 and 2). Genome recovery improvements were greatest for the high G+C strain *M. ruber* and least pronounced for the low G+C strain *P. marinus*, which confirmed earlier indications of diminishing efficiency of traditional MDA with increasing G+C of the template[28]. We speculate that this improvement may be caused by the higher reaction temperature of WGA-X (45 °C, as compared to 30 °C used in MDA) facilitating the priming, partial denaturing and dissociation from other cell constituents of high %GC DNA templates. WGA-X also outperformed MDA in genome recovery of two microalgae strains and led to the identification of regions that are missing in their reference genome assemblies (Fig. 1c, d; Supplementary Figs. 1 and 2; Supplementary Tables 1 and 2). Although SCG was used in several prior studies of unicellular eukaryotes[12, 40, 41], here we benchmarked this process, indicating that high quality de novo assemblies can be obtained from WGA-X SAGs of both prokaryotes and eukaryotes.

The application of WGA-X on individual, uncultured cells of environmental microorganisms confirmed improved genome

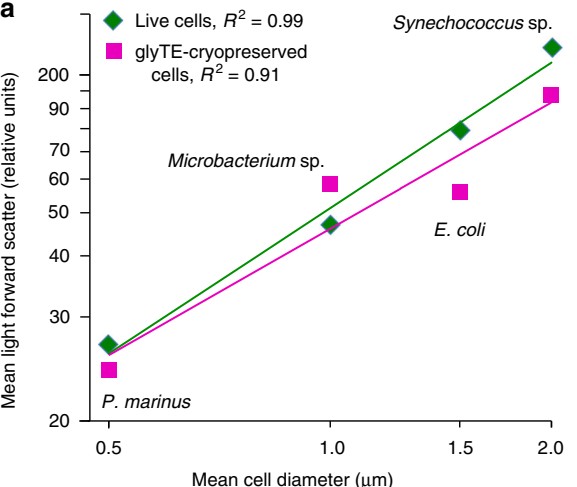
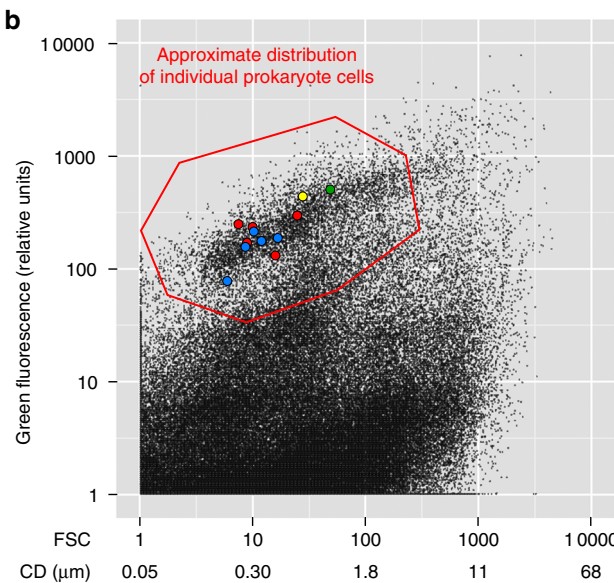
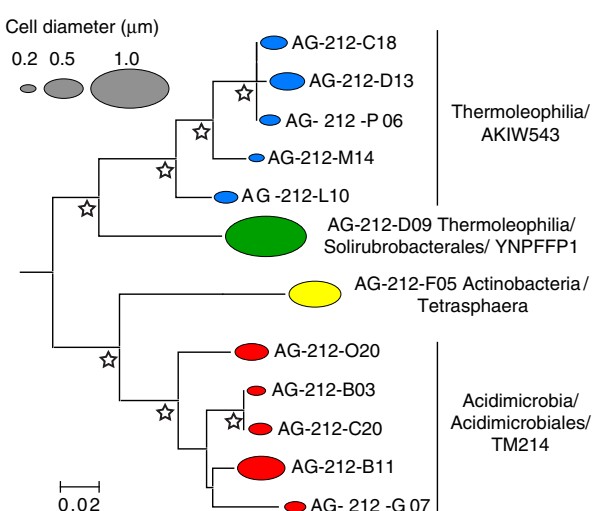

**Fig. 5** Cell diameter equivalent determination of soil prokaryote cells using calibrated index FACS. **a** Log-linear regression between light forward scatter and cell diameter of the following laboratory cultures, with approximate cell diameters in parenthesis: *P. marinus* (0.5 µm), *Microbacterium* sp. (1 µm), *E. coli* (1.5 µm), and *Synechococcus* sp. (2.0 µm). **b** Light forward scatter and green fluorescence of soil Actinobacteria cells stained with SYTO-9, in the context of other particles in the sample. CD = estimated cell diameter equivalents. *Colors* correspond to phylogenetic groups that are shown in **c**. **c** Estimated cell diameter equivalents and the SSU rRNA gene phylogeny of Actinobacteria SAGs. *Stars* indicate >80% bootstrap support

recovery, in particular for the soil microbiome that was dominated by high G+C genomes (Fig. 3). Our findings of G+C divergence between soil and marine microbiomes are in agreement with prior reports[42] and provide identities and genomic sequences of specific, yet uncultured taxa that contribute to the observed G+C distribution patterns. SCG studies of soil microbiomes would help accessing genomes of globally abundant, uncultured lineages[43]. We also show that traditional MDA is not well suited for SCG of soils and other microbiomes that are dominated by cells with high G+C content[42, 44], and that WGA-X effectively resolves this major technical challenge.

High-quality genome recovery from SAGs requires deep sequencing (Supplementary Fig. 5), which remains costly when applied on a large number of SAGs. Therefore, environmental SAGs are usually pre-selected using PCR-based sequencing of their SSU rRNA or other genes[1, 45]. This strategy, however, can be biased by primer mismatches[46], inserts in PCR templates[16], variable numbers of target gene copies per cell[47], or insufficient information contained in a PCR amplicon sequence. Improved SAG screening and more cost-effective genomic sequencing are of paramount importance to make SCG less prone to biases and more applicable in large-scale studies. Our results demonstrate LoCoS as a viable complement or even alternative to the PCR-based screening. LoCoS taxonomic assignments agreed with PCR-based identification when available (Supplementary Data 2), and with prior studies of similar environments[44, 48], giving no indications for taxonomic biases. LoCoS increased the fraction of identified prokaryote SAGs and detected multiple Chlamydiae in the soil sample, which were missed by the PCR of the SSU rRNA gene, likely due to known mismatches to the 27F PCR primer[46] (Figs. 2 and 3; Supplementary Data 1). In addition to the taxonomic assignment, LoCoS also enabled the analysis of G+C content (Fig. 3), partial metabolic pathways, and viral infections (Supplementary Data 2) in hundreds of individual cells of uncultured microorganisms in a single experiment.

SCG of infected cells (virocells) is a powerful tool for cultivation-independent studies of virus-host interactions in the environment[12, 17–19, 49, 50]. LoCoS offers scalability improvements for such analyses. In the two WGA-X and two MDA plates analyzed in this study, each containing 317 SAGs, LoCoS identified new phages in marine Alphaproteobacteria (including the ubiquitous SAR11 clade[51]), Gammaproteobacteria and Flavobacteria, as well as in four cells of soil bacteria (Supplementary Data 2). Notably, viral infections were detected in a twice larger fraction of WGA-X SAGs as compared to MDA SAGs, demonstrating the value of improved gDNA amplification in SCG-based studies of phage–host interactions. As DNA sequencing costs continue to decline, we expect that full-depth sequencing of all generated SAGs will become financially feasible, eliminating the need for pre-selection. LoCoS results offer a glimpse into these future opportunities, where virtually bias-free genomic data is obtained from individual cells and extracellular particles at a scale that adequately represents the genomic composition of complex microbiomes.

FACS is the most widely used cell separation and targeted sorting technique in microbial SCG due to its ability to utilize accurate measurement of multiple optical properties to automatically and rapidly separate target cells from other particles[1–4, 6]. However, standard FACS handles the collected optical data in a population mode, with no capacity to resolve the specific optical properties of those individual cells that undergo subsequent SCG analyses. The integration of index FACS into the microbial SCG workflow enabled a direct, high-throughput pairing of genomic and cell-size properties of individual cells and other particles (Figs. 4 and 5; Supplementary Fig. 3; Supplementary Data 1). For example, the exploration of marine

plankton (Fig. 4; Supplementary Fig. 3) confirmed prior suggestions that SAR11 dominates the low nucleic acid content cells[52], which is in agreement with their streamlined genome and metabolism[53].

The calibration of light forward scatter against cell diameters of a series of benchmark microbial cultures enabled us to determine diameter equivalents of those uncultured soil cells that were sorted for SCG. This approach revealed significant variation both between and within phylogenetic groups (Fig. 5; Supplementary Data 1). Assuming a spherical cell shape and the estimated average diameter equivalent of 0.65 µm (Supplementary Data 1), the average volume of bacterial cells in the analyzed soil sample was 0.14 µm³. This is on the low end of the average cell volume (0.16–0.30 µm³) determined microscopically for diverse soil types in an earlier study[54]. The finding of extremely small diameter equivalents of Thaumarchaea and Nitrospirae corroborates prior reports that the abundance of some prokaryote groups may be underestimated when collected on the commonly used 0.2 µm mesh-size filters[16]. Direct coupling of individual cell's size and its genome when analyzing uncultured microorganisms is important, as size is a major factor in cellular capacity to accumulate chemical constituents, perform metabolic functions, and interact with other cells in the environment. Until now, fluorescence in situ hybridization (FISH) targeting rRNA was the main technique allowing the examination of sizes and shapes of specific, uncultured microbial groups[55]. However, FISH is sensitive to the abundance and metabolic status (ribosome count) of target cells, is difficult to validate for false positives and false negatives in environmental samples, and requires the design of specific fluorescent probes for each target group. None of these limitations apply to the combination of index FACS and SCG, where the cell's physical properties are not tied to a pre-defined phylogenetic bin, enabling data processing at any phylogenetic granularity and direct linkage to whole-genome properties. In the future, this technique could be extended into diverse natural and induced fluorescence signals to link cellular genomic content and its chemical composition (e.g., Figs. 4 and 5; Supplementary Fig. 3), antigen presence[56], and specific metabolic activities[23].

Viral metagenomes are notoriously difficult to interpret due to high diversity and large numbers of unidentified genes[57]. Thus, the possibility of physical separation and subsequent sequencing of individual viral particles has been tested on cultured strains[58], and FACS has been employed to enrich for certain viral groups in targeted metagenomic studies[59]. Here, an untargeted exploration of extracellular particles in a marine sample, enabled by the combination of SCG and index FACS, recovered genomic sequences and optical properties of multiple relatives of previously cultured algal viruses and bacteriophages, as well as particles carrying DNA with no sequence homology in public databases (Fig. 4b; Supplementary Fig. 4; Supplementary Data 3). The latter, carried by particles of sub-cellular size, may represent novel viral groups, detrital DNA, or other types of extracellular genetic elements[60]. Notably, we report a 9× higher success rate in the de novo genome assembly of individual viral particles that were amplified by WGA-X, as compared to MDA (Supplementary Data 3). These results show that the FACS and DNA amplification and sequencing techniques developed for SCG are also applicable for the analysis of extracellular genetic elements, including phage-size particles, obtained directly from environmental samples. Such data can provide novel insights into the gene content, organization, and evolutionary histories of viruses and other extracellular particles in ways that already made a significant impact on our understanding of cellular life[1–6].

We see the combination of WGA-X, index FACS and LoCoS as a step forward toward new types of microbiome studies, where hundreds and thousands of discrete genomes (rather than

individual genes or metagenome bins) and phenotypic properties of uncultured single cells and extracellular genetic elements are accessed directly from environmental samples without major sampling biases, to address questions in biogeochemistry, ecology, evolution, and biotechnological potential. The improved genomic DNA amplification by WGA-X might also find applications beyond microbial SCG, including biomedical and forensic studies of human cells, such as tracking the genetic mosaicism related to cancer progression, neural disorders, and inherited abnormalities[4, 6].

## Methods

**Microbial samples**. Three bacterial cultures with a broad range of genomic G+C content, for which finished genomes are available in public databases, were employed in the optimization of WGA-X: *Prochlorococcus marinus*, *Eschericia coli*, and *Meiothermus ruber* (Supplementary Table 1). Cultures of two marine unicellular algae, *Thalassiosira pseudonana* and *Ostreococcus lucimarinus* (Supplementary Table 1) were also used in the subsequent benchmarking of optimized WGA-X and its comparison to MDA.

For marine prokaryote and extracellular particle analyses, the Gulf of Maine surface water was collected from 1 m depth in Boothbay Harbor, Maine (43°50′ 39.87″ N, 69°38′27.49″ W) on 15 June, 2011. One ml aliquots were amended with 5% glycerol and 1× TE buffer (all final concentrations), and stored at −80 °C until further analysis. The marine microalgae sample was collected from the same location on 16 September, 2009 and cryopreserved with 6% glycine betaine and 1× TE buffer (all final concentrations) at −80 °C. The soil sample was collected from 0–10 cm depth in a residential garden in Nobleboro, Maine (44°5′48.10″ N, 69°29′ 10.56″ W) on 5 May, 2015. Approximately 5 g of the soil sample was mixed with 30 ml sterile-filtered PBS, vortexed for 30 s at maximum speed, and centrifuged for 30 s at 2000 rpm (800×*g*). The obtained supernatant was used for cell sorting within 30 min, and processed as described above.

**Fluorescence-activated cell sorting (FACS)**. Prior to FACS, samples were diluted to below $10^5$ cell ml$^{-1}$ with filter-sterilized Sargasso Sea water (*P. marinus*, *T. pseudonana*, and *O. lucimarinus* cultures and marine samples) or 1× PBS (*E. coli* and *M. ruber* cultures) and pre-screened through a 40 μm (for bacteria and extracellular particles), or 70 μm (for eukaryotes) mesh size cell strainer (Becton Dickinson). Apart from cultured and environmental microalgae, all other samples were incubated with the SYTO-9 DNA stain (5 μM; Thermo Fisher Scientific) for 10–60 min. FACS was performed using a BD InFlux Mariner flow cytometer equipped with a 488 nm laser for excitation and either a 70 μm (for bacteria and extracellular particles) or 100 μm (for eukaryotes) nozzle orifice (Becton Dickinson, formerly Cytopeia). The cytometer was triggered on side scatter, and the "single-1 drop" mode was used for maximal sort purity. Gates for the sorting of microalgae were defined based on their orange and red auto-fluorescence. For all other cells, sort gate was defined based on particle green fluorescence (proxy to nucleic acid content), light side scatter (proxy to size), and the ratio of green vs. red fluorescence (for improved discrimination of cells from detrital particles). In addition, we performed sorting of a random subset of particles that fell outside the prokaryote sort gate in a marine sample. Cells and non-cellular particles were deposited into 384-well plates containing 600 nl per well of 1× TE buffer and stored at −80 °C until further processing. Of the 384 wells, 317 wells were dedicated for single particles, 64 wells were used as negative controls (no droplet deposition), and 3 wells received 10 particles each to serve as positive controls. The accuracy of droplet deposition into microplate wells was confirmed several times during each sort day, by sorting 3.46 μm diameter SPHERO Rainbow Fluorescent Particles (Sperotech Inc.) and microscopically examining their presence at the bottom of each well. In these examinations, <2% wells did not contain beads and <0.4% wells contained more than one bead. None of the sequenced SAGs appeared to have heterogeneous genetic material, providing further evidence for the accuracy of individual cell sorting. This is consistent with our prior findings, employing similar techniques[10, 17, 21, 29, 45, 61].

Index sort data was collected using the BD FACS Sortware software. The following laboratory cultures were used in the development of a cell diameter equivalent calibration curve: *Prochlorococcus marinus* CCMP 2389, *Microbacterium* sp., *Escherichia coli* K12 DH1, and *Synechococcus* CCMP 2515. Average cell diameters of these cultures were determined using an epifluorescence Axioskop microscope (Zeiss Inc.) equipped with a SPOT camera and software (Diagnostic Instruments Inc.). Cultures were cryopreserved with 5% glycerol and 1× TE buffer, and slides prepared by staining 1 ml sample with 4′,6-diamidino-2-phenylindole (DAPI, 5 μg ml$^{-1}$, final concentration) or SYTO-9 (5 μM, final concentration) and filtered onto black polycarbonate filters (pore size, 0.2 μm). Slides were examined using either 390–420 nm (violet) or 470–490 nm (blue) excitation. Cell images were acquired using a Zeiss 63X Plan-Neofluar objective (1.25 NA) yielding a resolution of ~0.1 μm per pixel. Cell cultures *P. marinus*, *Microbacterium* sp., *E. coli*, and *Synechococcus* sp. were found to be ~0.5, 1, 1.5, and 2 μm, respectively. Average light forward scatter of each of the four cultures was determined using the same BD InFlux Mariner settings as in environmental sample sorting and was repeated each day of single-cell sorting. In agreement with prior

reports[62, 63], we observed a strong correlation between cell diameters and light forward scatter (FSC) among these cultures (Fig. 4a). Taking advantage of this correlation, the diameter equivalent of the sorted environmental cells (*D*) was estimated from a log-linear regression model:

$$D = 10^{(a*\log10(\text{FSC})-b)},$$

where *a* and *b* are empirically derived regression coefficients (Fig. 4a).

**Cell lysis**. Prior to gDNA amplification, cells were lysed and their DNA was denatured by five freeze-thaw cycles (not applied on *P. marinus* and *E. coli*, based on prior experience), the addition of 700 nl of a lysis buffer consisting of 0.4 M KOH, 10 mM EDTA and 100 mM dithiothreitol, and a subsequent 10 min incubation at either 4 or 20 °C. The lysis was terminated by the addition of 700 nl of 1 M Tris-HCl, pH 4. A comparison of 10 min alkaline cell lysis at two temperatures demonstrated that faster gDNA amplification and better genome recovery were achieved from 20 °C lysates in the case of WGA-X and from 4 °C lysates in the case of MDA. These lysis conditions were used in subsequent, optimized reactions.

**Optimization and benchmarking of single-cell WGA-X**. A series of experiments were performed to optimize single-cell gDNA amplification reactions using the Equiphi29 polymerase (Thermo Fisher Scientific). We manipulated the following reaction conditions: polymerase concentration (0.4–4 U μl$^{-1}$), the length of random oligomers (six to eight nucleotides, with two 3′-terminal nucleotide bonds phosphorothioated), oligomer concentration (1–50 μM), reaction buffer composition, and reaction temperature (40–45 °C). Our earlier work indicated that the speed of single-cell MDA reactions, measured as reaction's critical point (Cp; Fig. 1b), correlates with the fraction of the genome that can be recovered from the obtained amplicons[50]. Therefore, our initial efforts were geared to achieve the maximal difference between the Cp of no-template (negative control) reactions vs. reactions containing one cell. The genomic sequencing, de novo assembly and assembly QC of selected SAGs were used to verify Cp-based findings, and to guide further optimization to identify gDNA amplification conditions that produce longest and most contiguous genome assemblies with fewest artifacts (bases that do not align to references, misassemblies, mismatches, and indels). The optimized, 10 μl WGA-X reactions contained 0.2 U μl$^{-1}$ Equiphi29 polymerase (Thermo Fisher Scientific)[38], 1× Equiphi29 reaction buffer (Thermo Fisher Scientific), 0.4 mM each dNTP (New England BioLabs), 10 mM dithiothreitol (Thermo Fisher Scientific), 40 μM random heptamers with two 3′-terminal phosphorothioated nucleotide bonds (Integrated DNA Technologies), and 1 μM SYTO-9 (Thermo Fisher Scientific) (all final concentrations). These reactions were performed at 45 °C for 12–16 h, then inactivated by a 15 min incubation at 75 °C.

The performance of the optimized WGA-X was evaluated by comparing the quality of WGA-X-based benchmark culture SAG genome assemblies to the assemblies of SAGs of the same cultures that were obtained using standard MDA reactions. The 10 μl MDA reactions contained 0.1 U μl$^{-1}$ of phi29 polymerase (New England BioLabs), 1× reaction buffer (New England BioLabs), 0.4 μM each dNTP (New England BioLabs), 10 μM dithiothreitol (Thermo Fisher Scientific), 50 μM random hexamers with two 3′-terminal phosphorothioated nucleotide bonds (Integrated DNA Technologies) and 1 μM SYTO-9 (Thermo Fisher Scientific) (all final concentrations). The MDA reactions were run at 30 °C for 12–16 h, then inactivated by a 15 min incubation at 65 °C. The kinetics of all MDA and WGA-X reactions were monitored by measuring SYTO-9 fluorescence with a FLUOstar Omega (BMG) plate reader. The Cp was determined for each reaction as the time required to produce half of the fluorescence after reaching a plateau (Fig. 1b). Amplified genomic DNA from WGA-X and MDA reactions was stored at −80 °C until further processing.

To prevent WGA-X and MDA reactions from contamination with non-target DNA, all cell lysis and gDNA amplification reagents were treated with UV in a Stratalinker (Stratagene)[64]. An empirical optimization of the UV exposure was performed to determine the length of UV exposure that is necessary to crosslink all detectable contaminants without inactivating the reaction. To further reduce the risk of SAG contamination with DNA in WGA-X reagents, we generated a contaminant sequence database and implemented a computational filter to remove similar sequences from SAG reads and assemblies. First, WGA-X products were generated in 384-well, no-template, 10 μl reactions as previously described, but without applying UV decontamination. Products of these reactions were sequenced using LoCoS procedures and co-assembled following same protocols as described above. This resulted in a 3.2 Mbp co-assembly (MG-RAST #4732992.3). No 16S rRNA gene sequences were retrieved in this co-assembly. The MG-RAST default annotation indicated a diverse taxonomic composition, with the predominance of Proteobacteria. We utilized these contaminant sequences as a reference database to identify and remove raw reads and contigs containing >100 bp regions with >95% nucleotide identity in subsequently sequenced microbial SAGs, using Borrows Wheel Aligner (BWA)[65] and BLASTn[66], respectively.

Cell sorting, lysis, and gDNA amplification setup were performed in a HEPA-filtered environment conforming to Class 1000 cleanroom specifications. Prior to cell sorting, the instrument, the reagents, and the workspace were decontaminated for DNA using UV irradiation and sodium hypochlorite solution, as previously described[61]. To further reduce the risk of DNA contamination, and to improve accuracy and throughput, Bravo (Agilent Technologies) and Freedom Evo (Tecan) robotic liquid handlers were used for all liquid handling in 384-well plates.

# ARTICLE

**Genomic sequencing of microbial SAGs.** Libraries for SAG genomic sequencing were created with Nextera XT (Illumina) reagents following manufacturer's instructions, except for purification steps, which were done with column cleanup kits (QIAGEN), and library size selection, which was done with BluePippin (Sage Science, Beverly, MA, USA), with a target size of 500 ± 50 bp. DNA concentration measurements were performed with Quant-iT™ dsDNA Assay Kits (Thermo Fisher Scientific) and a FLUOstar Omega (BMG) plate reader, following manufacturer's instructions. Libraries were sequenced with NextSeq 500 (Illumina) in 2 × 150 bp mode using v.1 reagents. The obtained sequence reads were quality-trimmed with Trimmomatic v0.32[67] using the following settings: -phred33 LEADING:0 TRAILING:5 SLIDINGWINDOW:4:15 MINLEN:36. Human DNA (≥95% identity to the *H. sapiens* reference assembly GRCh38) and low complexity reads (containing <5% of any nucleotide) were removed. The reads were aligned to their respective reference genomes using BWA[65]. Basic mapping and chimera statistics were obtained using SAMtools[68] on the mapped reads. Genome coverage was calculated with the aligned reads using BEDTools[69]. Variants across the aligned reads were called using FreeBayes[70]. For de novo assemblies, quality-filtered reads (as above) were digitally normalized with kmernorm 1.05 (http://sourceforge.net/projects/kmernorm) using settings -k 21 -t 30 -c 3 and then assembled with SPAdes v.3.0.0 using the following settings: --careful --sc --phred-offset 33. Each end of the obtained contigs was trimmed by 100 bp, and then only contigs longer than 2000 bp were retained. To determine the optimal sequencing depth for bacterial and algal benchmark SAGs, reads were randomly down-sampled to between 0.2 and 5, and 5 and 20 million reads per SAG, respectively. A rarefaction analysis suggested that sequencing more than five million paired-end reads per bacterial SAG and 20 million paired-end reads per algal SAG did not introduce substantial further improvements in genome recovery (Supplementary Fig. 5). Therefore, all genomic comparisons of benchmark SAGs were performed using assemblies that were produced from 5 million reads in the case of bacteria and 20 million reads in the case of algae.

Assembly quality of benchmark SAGs was evaluated with QUAST[71]. The phylogenetic assignments of prokaryote assemblies were performed by CheckM[72] on conserved protein-coding genes and CREST[73] on SSU rRNA sequences retrieved with Metaxa2[74]. Viral contigs were identified using a combination of searches for viral marker genes, G+C content, tetramer frequency anomalies, and elevated fragment recruitment of viral vs. bacterial metagenomic reads, as previously described[17]. Whole-genome synteny comparisons were performed with EasyFig for Mac version 2.1[75] using tBLASTx with the filtering of small hits and annotations option.

Low coverage sequencing (LoCoS) was performed on environmental SAGs as a cost-effective means to obtain minimal genomic data from a maximal number of individual cells. Uniquely barcoded Illumina libraries were generated as described above for each well on a 384-well microplate. These libraries were co-sequenced, without pre-normalization, in a single NextSeq 500 run using a medium-output kit. This sequencing effort produced an average of 291,411 paired-end reads per well, with a range of 0–8,389,480 reads per well. The de novo assemblies and phylogenomic characterization of LoCoS reads were performed using the same protocols as the full-depth SAG sequencing.

**PCR-based SAG identification.** WGA-X and MDA amplification products were diluted 50-fold in TE buffer to serve as templates in PCR reactions targeting the small subunit (SSU) rRNA gene. The reactions were performed using LightCycler 480 SYBR Green I Master Mix (Roche) and a LightCycler® 480 II real-time thermal cycler (Roche) using domain-specific primers 27F/907R (Bacteria), 344F/915R (Archaea), and Euk528F/EukB (Eukaryotes) and previously described thermal conditions[45, 76]. Products of PCR reactions that were significantly faster that PCR reactions on no-template WGA-X or MDA controls were sequenced from both ends using Sanger technology at GENEWIZ (Cambridge, MA, USA). The two reads were aligned and manually curated with Sequencher (Gene Codes). The resulting consensus sequences were classified with CREST[73] (Bacteria and Archaea) and Silva Incremental Aligner[77] (eukaryotes).

**Data availability.** DNA sequence data reported in this manuscript are publicly accessible through MG-RAST (http://metagenomics.anl.gov/). The accession numbers are: 290640 (SSU rRNA genes from marine prokaryotes), 290634 (SSU rRNA genes from soil prokaryotes), 290633 (SSU rRNA gene from microalgae), 290644 (LoCoS assemblies from marine prokaryotes amplified with MDA), 290642 (LoCoS assemblies from marine prokaryotes amplified with WGA-X), 290636 (LoCoS assemblies from soil prokaryotes amplified with MDA), 290632 (LoCoS assemblies from soil prokaryotes amplified with WGA-X), 290638 (LoCoS assemblies from marine extracellular particles amplified with WGA-X), 290635 (full depth WGS assemblies from marine extracellular particles amplified with MDA), 290631 (full depth WGS assemblies from marine extracellular particles amplified with WGA-X), and 4732992.3 (assembly of DNA contaminants in WGA-X reagent without UV treatment). The authors declare that all other relevant data supporting the findings of the study are available within this published article and its Supplementary Information files, or from the corresponding author upon request.

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

## Acknowledgements

We are thankful to Drs Ger van den Engh, Barclay Purcell, and the BD Corporation for their help with interpreting the index sort data obtained from the BD Influx Sortware software. We thank Dr David Emerson for a critical read of the manuscript, Dr. Pavel Pevzner for advice on the use of SPAdes assembly software, and the Illumina Corporation for the MiSeq award to Dr Ramunas Stepanauskas. Financial support for this study was provided by the Bigelow Laboratory for Ocean Sciences, the U.S. National Science Foundation grants OCE-1335810 and DEB-1441717 (to R.S.) and the National Aeronautics and Space Administration Grant NNX15AM11G (to R.S.).

## Author contributions

R.S.: experiment design, project management, data analysis and visualization, and manuscript preparation. E.A.F.: WGA-X development. J.B. and J.M.B.: development and implementation of computational tools for de novo genome assembly and quality control. N.J.P., B.T., and B.P.T.: index FACS development and implementation. T.P. and A.L: Equiphi29 polymerase development. J.M.L.: viral sequence analyses. E.D.B. and M.G.P.: environmental microbiome data analyses. C.J.M. and W.K.B. single-cell genomics laboratory analyses. All contributions toward manuscript preparation.

## Additional information

**Competing interests:** A.L. and T.P. work for Thermo Fisher Scientific, which markets several reagents that were used in this study. R.S., E.A.F., J.B., N.J.P., B.T., J.M.L., E.D.B, J.M.B., M.G.P., B.P.T., C.J.M., and W.K.B. are employees of Bigelow Laboratory for Ocean Sciences, which provides per-fee core facility services in single-cell genomics.

**Change history:** A correction to this article has been published and is linked from the HTML version of this paper.

