## [Peer Review File · Nature Communications]

Reviewers' comments:

Reviewer #1 (Remarks to the Author):

This paper reports single-cell testing of a recently published phi29 polymerase enzyme variant (ref 33) with a focus on improved performance for GC-rich bacteria. The paper also touts "index sorting" as a major advance. While this represents a new way to use a flow cytometer, it "index sorting" provides no new information compared with standard gated sorting, which would allow the same cell size and DNA content calibrations/determinations to be made. This manuscript would be stronger if the "index sorting" part were removed and any slightly improved resolution of optical signals over standard FACS developed separately.

The work with high-GC uncultivable bacteria is sound and the improved performance on high GC bacteria significant. However, a key comparison is missing – that to the REPLI-g SensiPhi DNA Polymerase, another enhanced Phi29 polymerase for MDA available from Qiagen. The academic work from the Salas lab (de Vega et al, PNAS, 2010 and others) underlying the Qiagen product is not cited. Even with an added comparison to SensiPhi, I would still be unsure about the suitability of this manuscript for the general audience at Nature Communications – maybe the authors could test the performance of WGA-X versus WGA chemistries popular with biomedical researchers like MALBAC and PicoPlex on human cells to make the manuscript more interesting to cancer researchers. It would also be helpful if the authors, as experts on single-cell genomics methods, could recommend when users should select regular MDA, the SensiPhi polymerase, the WGA-X polymerase, and/or small liquid volume methods, at least for bacterial samples.

Reviewer #2 (Remarks to the Author):

Review comments on

Manuscript # Nature Communications NCOMMS-17-04689-T

Title "Improved genome recovery and integrated phenotype analyses of individual, uncultured microbial cells and viral particles" by Dr Stepanauskas and coworkers.

This manuscript is a technical note, describing WGA-X method to enhance genome recovery from individual microbial cells. New developed WGA-X was still based on MDA but modified reaction temperature, random primer length and reaction buffer. These changes were key to WGA-X. The authors randomly took various samples to confirm WGA-X has significantly improved genome coverage of single cells.

Main concerns:

1. Over last 3 years, many papers have demonstrated that data analysis alone such as genome binning was able to get single cell genomes with high genomic coverage from metagenomes, metagenomic analysis can predict and link species and functions, and most bacteria in human gut are culturable. How useful and urgent is this SCG? It would be helpful to discuss the issue.
2. I agree with the statement that it is important to integrate SCG with single cell phenotype analyses. However, it is a little misleading that the phenotype in this manuscript refers only sizes of cells: 'we were able to analyze both genomes and phenotype properties, including sizes...' Bacterial size and shape vary significantly which are hard to be used as useful phenotype. Besides, a common microscope can also observe sizes and shape of cells. There is no improvement compared to other traditional technology if phenotype means sizes of cells here.
3. 'To the best of our knowledge, this is the first SCG study of a soil microbiome,...'. To perform SCG from soil, it is critical to extract and recover most cells from soil. This manuscript used a

simple PBS and vortex extraction, is this method efficient enough to get representative soil bacteria?

4. In this manuscript, all seawater samples were stained with SYTO-9, would this DNA stain affect later MDA and WGA-X process?

5. Why 1 μL SYTO-9 was added into 10 μL WGA-X reaction solution? Is it only used to monitor the DNA concentration? would SYTO-9 be necessary step? would it affect subsequent DNA sequencing?

Responses to Reviewers

Reviewer #1

This paper reports single-cell testing of a recently published phi29 polymerase enzyme variant (ref 33) with a focus on improved performance for GC-rich bacteria. The paper also touts “index sorting” as a major advance. While this represents a new way to use a flow cytometer, it “index sorting” provides no new information compared with standard gated sorting, which would allow the same cell size and DNA content calibrations/determinations to be made. This manuscript would be stronger if the “index sorting” part were removed and any slightly improved resolution of optical signals over standard FACS developed separately.

> We agree with the reviewer that improved single cell DNA amplification is the most significant advance reported in our manuscript. However, we believe that the integration of single cell genomics with calibrated index FACS also provides a major scientific advance. The reviewer’s comment about targeted gating providing similar information may be true for work on pure microbial cultures or other low-diversity samples. However, in most natural microbial communities, where thousands of diverse lineages co-exist, gating each of them separately is not a feasible option. Uncultured lineages constitute over 99% of cells in most environments. Cell size data is not available for these predominant organisms and their genomic sequences are of limited use predicting such phenotypes at this point. We have included relevant rationale and a comparison of our approach to other, commonly used techniques in environmental microbiology on lines 264-286.

The work with high-GC uncultivable bacteria is sound and the improved performance on high GC bacteria significant. However, a key comparison is missing – that to the REPLI-g SensiPhi DNA Polymerase, another enhanced Phi29 polymerase for MDA available from Qiagen. The academic work from the Salas lab (de Vega et al, PNAS, 2010 and others) underlying the Qiagen product is not cited. Even with an added comparison to SensiPhi, I would still be unsure about the suitability of this manuscript for the general audience at Nature Communications – maybe the authors could test the performance of WGA-X versus WGA chemistries popular with biomedical researchers like MALBAC and PicoPlex on human cells to make the manuscript more interesting to cancer researchers. It would also be helpful if the authors, as experts on single-cell genomics methods, could recommend when users should select regular MDA, the SensiPhi polymerase, the WGA-X polymerase, and/or small liquid volume methods, at least for bacterial samples.

> We appreciate the suggestion to cite Vega et al. 2011, which is now done on line 63. We reviewed the alternative strategies for microbial single cell whole genome amplification on lines 52-71. MDA has been used in the vast majority of microbial single cell genomics studies to date, with no publication suggesting that other methods outperform MDA in this particular application. This is also true for the protein-primed amplification described in Vega et al. 2011. It would not be feasible for us to experimentally compare all previously published WGA iterations, some of which are not even compatible with other components of our integrated workflow, e.g. index FACS, robotic liquid handling, high-throughput genomic sequencing library preparation, and de novo genome assembly.

Reviewer #2

This manuscript is a technical note, describing WGA-X method to enhance genome recovery from individual microbial cells. New developed WGA-X was still based on MDA but modified reaction temperature, random primer length and reaction buffer. These changes were key to WGA-X. The authors randomly took various samples to confirm WGA-X has significantly improved genome coverage of single cells.

Main concerns:

1. Over last 3 years, many papers have demonstrated that data analysis alone such as genome binning was able to get single cell genomes with high genomic coverage from metagenomes, metagenomic analysis can predict and link species and functions, and most bacteria in human gut are culturable. How useful and urgent is this SCG? It would be helpful to discuss the issue.

> While metagenomic assembly and binning has many powerful applications, it does not provide genomes from single cells. Metagenome assembly bins are mosaics of many individuals, often containing genomic heterogeneity in operationally defined microbial populations. We also respectfully disagree with the reviewer regarding the cultivability of most environmental microorganisms. In fact, we are not aware of a single microbiologist who would question the fact that the vast majority of microorganisms in marine and soil environments, which we analyzed in this study, remain uncultured. The fraction of cultured lineages in human guts is higher, but still remains well below 50%. To address reviewer's comment, we provide the rationale for microbial single cell genomics on lines 31-47: "Single cell genomics (SCG) is a transformative research tool that retrieves information-rich genomic blueprints from the most fundamental units of biology. This is particularly significant in the case of bacteria, archaea and protists, where individual cells constitute complete organisms. Such unicellular individuals comprise the vast majority of biological diversity on our planet, with recent estimates indicating over a trillion species. Only a small fraction of them are amenable to the cultivation-based, classical microbiology studies. SCG, along with other cultivation-independent research tools, is instrumental in the deciphering of the biological features of many deep branches of the tree of life that constitute a significant fraction of our planets biota yet remained unknown to science until recently. In addition, due to its ability to retrieve genetic information from all DNA molecules in a cell, SCG opens a window of opportunity to study microbial physical interactions, such as infections, symbioses and predation, directly in their natural environment. Finally, by circumventing the need for arbitrary taxonomic binning, SCG improves our understanding of microbial microevolutionary processes and helps calibrating the performance and interpretation of community omics tools. A major, still underutilized opportunity lies in the integration of SCG with single cell phenotype analyses, which can provide deeper insights into the roles of uncultured microbial groups in nature and inform their practical utilization in biotechnology".

2. I agree with the statement that it is important to integrate SCG with single cell phenotype analyses. However, it is a little misleading that the phenotype in this manuscript refers only sizes of cells: 'we were able to analyze both genomes and phenotype properties, including sizes...' Bacterial size and shape vary significantly which are hard to be used as useful phenotype. Besides, a common microscope can also observe sizes and shape of cells. There is no improvement compared to other traditional technology if phenotype means sizes of cells here.

> The key advantage of the described technology is its capacity to connect genomic and phenotypic properties of the uncultured microbial groups. This cannot be achieved by microscopy. Although in this particular study we analyzed only cell sizes and the fluorescence of nucleic acid probes, the same approach can be easily applied to study other phenotypic properties, as discussed on lines 283-286. A detailed discussion of the advantages of SCG-index FACS integration is provided on lines 264-286. In response to reviewer's suggestion, we have replaced the term "phenotypic properties" with the more specific term "cell size" in the Abstract, line 25. Cell size was the primary phenotype feature analyzed in our study. However, we also demonstrate the potential for this approach to be expanded into other phenotypic properties, such as the nucleic acid content (lines 169-171, 261-263) and metabolic activity (lines 283-286).

3. 'To the best of our knowledge, this is the first SCG study of a soil microbiome,...'. To perform SCG from soil, it is critical to extract and recover most cells from soil. This manuscript used a simple PBS and vortex extraction, is this method efficient enough to get representative soil bacteria?

> No perfect method exists for an unbiased extraction of cells or DNA from soil, and we make no claim for achieving such a goal in our manuscript. What we claim is that our approach is well suited to obtain genomic information from some of the abundant but as of yet uncultured soil microbial lineages. The value of this approach for soil microbiomes has already been demonstrated in a prior study, reference #39.

4. In this manuscript, all seawater samples were stained with SYTO-9, would this DNA stain affect later MDA and WGA-X process?

> A DNA stain is needed to differentiate cells from other particles of similar size when analyzing environmental samples by FACS. We chose SYTO-9, due to its low affinity to DNA (manufacturer's information), thus minimizing potential interferences with polymerases. This approach produces good genome recovery from single cells, as demonstrated in this manuscript and in over 60 prior, peer-reviewed publications that employed the same staining technique since 2006.

5. Why 1 μ L SYTO-9 was added into 10 μ L WGA-X reaction solution? Is it only used to monitor the DNA concentration? would SYTO-9 be necessary step? would it affect subsequent DNA sequencing?

> SYTO-9 was added in order to stain amplified DNA and in this way monitor the kinetics of each WGA-X and MDA reaction, as described on lines 413-417. No interferences with downstream sequencing were observed in this study, nor in any of our prior studies going back to 2006.

REVIEWERS' COMMENTS:

Reviewer #2 (Remarks to the Author):

Further comments on the revised manuscript:

1. The title using 'Improved genome recovery and integrated phenotype analyses' is a little bit misleading. In the manuscript, a lot of statements claim this work link genome and phenotype. However, this work can only measure size of cells which is very limited information of 'phenotype'. Furthermore, size measurement is not reliable to define a phenotype, it is well known that the same species at different physiological state has different size and shape. A true phenotype has much more meaning than the size of cells. It would be better to modify the manuscript to reflect the size measurement, instead of claiming phenotype.

2. I fully understand and appreciate that the study of uncultured bacteria in nature is very important. What I argue is advanced metagenomic analysis could be powerful enough to circumvent single cell genomics and reconstruct genomes of cells which are difficult to cultivate, for example two binning work described in Nature Biotechnology 32, 822–828 (2014) and Nature Biotechnology 31, 533–538 (2013). It would be good to modify the manuscript reflecting these binning analysis.

3. Metagenomic analysis is able to sufficiently link phylogeny and function without single cell genomics and this 'predictive metagenomic' approach has provided insights into the thousands of uncultivated microbial communities (Nature Biotechnology 31, 814–821 (2013)). Authors should also mention and comment this work in the manuscript and justify this study.

Responses to Reviewers

Reviewer #2

Further comments on the revised manuscript:

1. The title using 'Improved genome recovery and integrated phenotype analyses' is a little bit misleading. In the manuscript, a lot of statements claim this work link genome and phenotype. However, this work can only measure size of cells which is very limited information of 'phenotype'. Furthermore, size measurement is not reliable to define a phenotype, it is well known that the same species at different physiological state has different size and shape. A true phenotype has much more meaning than the size of cells. It would be better to modify the manuscript to reflect the size measurement, instead of claiming phenotype.

> Although we present qualitative data for multiple phenotypic properties of the analyzed cells (e.g. DNA and chlorophyll fluorescence, light scatter; see Figures 4 and 5, Supplementary Figure 3), we agree with the reviewer that our quantitative data is limited to cell size. We have modified the manuscript title and text accordingly.

2. I fully understand and appreciate that the study of uncultured bacteria in nature is very important. What I argue is advanced metagenomic analysis could be powerful enough to circumvent single cell genomics and reconstruct genomes of cells which are difficult to cultivate, for example two binning work described in Nature Biotechnology 32, 822–828 (2014) and Nature Biotechnology 31, 533–538 (2013). It would be good to modify the manuscript reflecting these binning analysis.

3. Metagenomic analysis is able to sufficiently link phylogeny and function without single cell genomics and this 'predictive metagenomic' approach has provided insights into the thousands of uncultivated microbial communities (Nature Biotechnology 31, 814–821 (2013)). Authors should also mention and comment this work in the manuscript and justify this study.

> We agree with the reviewer that metagenomics is an instrumental approach in studies of uncultured microorganisms and therefore included citations to the suggested publications.